# Pathophysiological and Pharmacological Characteristics of *KCNJ5* 157-159delITE Somatic Mutation in Aldosterone-Producing Adenomas

**DOI:** 10.3390/biomedicines9081026

**Published:** 2021-08-17

**Authors:** Kang-Yung Peng, Hung-Wei Liao, Jeff S. Chueh, Chien-Yuan Pan, Yen-Hung Lin, Yung-Ming Chen, Peng-Ying Chen, Chun-Lin Huang, Vin-Cent Wu

**Affiliations:** 1Department of Internal Medicine, National Taiwan University Hospital, Taipei 100, Taiwan; pengky68@gmail.com (K.-Y.P.); austinr34@gmail.com (Y.-H.L.); chenym@ntuh.gov.tw (Y.-M.C.); becky60523becky@gmail.com (P.-Y.C.); a0955726792@gmail.com (C.-L.H.); 2Chinru Clinic, Taipei 116, Taiwan; lhw898@gmail.com; 3Department of Urology, College of Medicine, National Taiwan University, and National Taiwan University Hospital, Taipei 110, Taiwan; jeffchueh@gmail.com; 4Department of Life Science, College of Life Science, National Taiwan University, Taipei 106, Taiwan; cypan@ntu.edu.tw

**Keywords:** Aldosterone producing adenoma, *KCNJ5*, 157-159delITE mutation, primary aldosteronism

## Abstract

Mutated channelopathy could play important roles in the pathogenesis of aldosterone-producing adenoma (APA). In this study, we identified a somatic mutation, *KCNJ5* 157-159delITE, and reported its immunohistological, pathophysiological and pharmacological characteristics. We conducted patch-clamp experiments on HEK293T cells and experiments on expression of aldosterone synthase (CYP11B2) and aldosterone secretion in HAC15 cells to evaluate electrophysiological and functional properties of this mutated *KCNJ5*. Immunohistochemistry was conducted to identify expressions of several steroidogenic enzymes. Macrolide antibiotics and a calcium channel blocker were administrated to evaluate the functional attenuation of mutated *KCNJ5* channel in transfected HAC15 cells. The interaction between macrolides and KCNJ5 protein was evaluated via molecular docking and molecular dynamics simulation analysis. The immunohistochemistry analysis showed strong CYP11B2 immunoreactivity in the APA harboring *KCNJ5* 157-159delITE mutation. Whole-cell patch-clamp data revealed that mutated *KCNJ5* 157-159delITE channel exhibited loss of potassium ion selectivity. The mutant-transfected HAC15 cells increased the expression of CYP11B2 and aldosterone secretion, which was partially suppressed by clarithromycin and nifedipine but not roxithromycin treatment. The docking analysis and molecular dynamics simulation disclosed that roxithromycin had strong interaction with *KCNJ5* L168R mutant channel but not with this *KCNJ5* 157-159delITE mutant channel. We showed comprehensive evaluations of the *KCNJ5* 157-159delITE mutation which revealed that it disrupted potassium channel selectivity and aggravated autonomous aldosterone production. We further demonstrated that macrolide antibiotics, roxithromycin, could not interfere the aberrant electrophysiological properties and gain-of-function aldosterone secretion induced by *KCNJ5* 157-159delITE mutation.

## 1. Introduction

Primary aldosteronism (PA) is the most common form of secondary hypertension, characterized by elevated plasma aldosterone and low renin hypertension, and affects ~10% of hypertensive patients [1,2]. These patients have a higher rate of cardiovascular consequences than that of essential hypertensive (EH) patients [1,2].

The pathogenesis of aldosterone producing adenoma (APA) is attributed to the over-proliferation of adrenal cortical cells and aberrant aldosterone production [3]. Recent studies indicate that somatic mutations of the potassium channel *KCNJ5* gene could be identified in 34 to 73% of APAs [4,5,6,7]. Additionally, other ion channel mutations have also been reported to involve in the pathogenesis of APA including loss-of-function mutations of *ATP1A1* and *ATP2B3* genes, and gain-of-function mutation of *CACNA1D* gene, which are present in 1 to 8% of APAs [8,9,10].

*KCNJ5* encodes the inwardly rectifying potassium channel Kir3.4 that exists both as homotetramers and heterotetramers with Kir3.1 (encoded by *KCNJ3*) [11]. Using electrophysiologic studies, it has been clearly demonstrated that *KCNJ5* mutations result in loss of channel selectivity leading to membrane depolarization. *KCNJ5* G151R and L168R are the most common somatic mutations in sporadic APAs in previous studies [4,12]. These two *KCNJ5* mutations are located in or near the selectivity filter in the glycine–tyrosine–glycine (GYG) motif of the Kir3.4 protein [4,13]. It has been suggested that the mutations around the GYG motif in KCNJ5 cause a loss in potassium selectivity and an increase in sodium influx into cytoplasm, resulting in the depolarization of the plasma membrane, thereby activating the voltage-gated calcium channels and downstream signaling pathways; then leading to increased aldosterone secretion [14,15]. Macrolide antibiotics, such as roxithromycin and clarithromycin, have been shown to blunt mutant KNCJ5 channels and consequently to inhibit CYP11B2 expression and aldosterone production [16]. Using this characteristic, macrolide has the potential to diagnose *KCNJ5*-mutant PA before adrenalectomy [17]. In this study, we identified a *KCNJ5* 157-159delITE mutation, which is located near the selectivity filter of the potassium channel and functionally similar to the previously reported G151R and L168R substitution mutations. We also evaluated whether macrolide drugs can inhibit the hyperaldosteronism function of this mutant *KCNJ5* channel in in vitro experiments.

## 2. Materials and Methods

### 2.1. Unilateral PA (uPA) Identification

Screening, confirmation, and the subtype identification of incident PA patients were performed in referred hypertensive patients according to the standard TAIPAI protocol and aldosteronism consensus in Taiwan [18]. All original anti-hypertensive medications were discontinued for at least 21 days before PA screening and confirmatory tests. Doxazosin and/or diltiazem were administered to control markedly high blood pressure (BP) during the work-up stage when required. The diagnosis of PA in hypertensive patients was based on the inappropriate hypersecretion of aldosterone and according to the fulfillment of the standard criteria [19] (methods detailed in the Appendix A).

Ethical approval (approval number 200611031R) was obtained from the institutional review board of the National Taiwan University Hospital. Written informed consent for clinical data collection, including genomic analysis and research use was obtained from this patient.

### 2.2. Nucleic Acid Extraction

Genomic DNA was extracted from adrenal tumor and peripheral whole blood. Tumor genomic DNA was extracted using QIAamp DNA mini kit (Qiagen, Hilden, Germany); genomic DNA from whole blood was extracted using Blood DNA Isolation Kit (GEB100/GEB01K, Geneaid Biotech; New Taipei City, Taiwan) according to the manufacturer’s instructions.

### 2.3. Sequencing of the KCNJ5 Gene

Coding sequence of genomic DNA was investigated by exome sequencing. The coding sequences and the intron-exon junctions of the genes *KCNJ5* were amplified by PCR and bidirectionally sequenced using the BigDye^®^ Terminator v3.1 Cycle Sequencing Kit (Applied Biosystems Inc., Foster City, CA, USA) with a 3730 DNA Analyzer (Applied Biosystems, Foster City, CA, USA). The primers used for the PCR amplification of the genomic DNA were designed using the exon/intron junction sequences as previous reported [20]. Sequences were analyzed using DNAStar Lasergene SeqMan Pro 7.1.0 software (DNAStar Inc., Madison, WI, USA).

### 2.4. Tissue Immunohistochemistry

Immunohistochemistry (IHC) was conducted using mouse monoclonal antibody for CYP11B2 and 17α-hydroxylase (CYP17A1), rat monoclonal antibody for CYP11B1 (generous gifts from Professor Celso Gomez-Sanchez [21]), and mouse monoclonal antibody for HSD3B (RRID:AB_425493, Abnova, Taipei, Taiwan). Sections of paraffin-embedded adrenal tumor and surrounding tissues were stained using the non-biotin-amplified method (Novolink; Novocastra Laboratories Ltd., Newcastle Upon Tyne, UK) according to the manufacturers’ protocol. Images were acquired using Olympus BX51 fluorescence microscope with a built-in Olympus DP72 camera and processed using cell Sens Standard 1.14 software (Olympus, Hamburg, Germany).

### 2.5. Cell Culture

HEK293T cells are cultured in DMEM (Gibco; Thermo Fisher Scientific, Waltham, MA, USA) supplemented with 10% fetal bovine serum (FBS; Gibco; Thermo Fisher Scientific) and antibiotics at 37 °C and 5% CO_2_. HAC15 human adrenocortical cells [22] are cultured in HAC15 complete media containing DMEM:F12 (1:1) (Gibco; Thermo Fisher Scientific, Waltham, MA, USA) supplemented with 10% FBS, 1X ITS (Sigma-Aldrich, St. Louis, MO, USA), 1% penicillin–streptomycin (Gibco; Thermo Fisher Scientific, Waltham, MA, USA) at 37 °C under an atmosphere and maintained at 37 °C in a humidified 5% CO_2_ incubator as previously reported [23,24].

### 2.6. Mutant KCNJ5 Transfection and Drugs Treatment

The plasmids expressing the wild-type *KCNJ5* as well as *KCNJ5* 157-159delITE mutation were constructed into the pIRES-EGFP-puro vector (Addgene plasmid #45567) using PCR-assisted, site-directed mutagenesis. The successful introduction of these mutations into the wild-type construct was confirmed by a PCR-based direct sequencing method. Human full-length cDNA of *KCNJ3* (SC118769; Origene Rockville, MD, USA) was constructed into pTagRFP-N vector.

To evaluate the effect of mutant *KCNJ5* on expression of CYP11B2 and aldosterone secretion, the HAC15 cells were seeded into 6-well plates (Corning 3516) at a density of 1×10^6^ cells/well. After 48 h, the medium was replaced with 1 mL Opti-MEM medium (Gibco; Thermo Fisher Scientific, Waltham, MA, USA) in each well. For each well of cells to be transfected, a total of 4 μg of pIRES-GFP empty vector, wild-type or mutant *KCNJ5* plasmid DNA in 250 μL Opti-MEM medium was mixed with 3.5 μL Lipofectamine LTX and 4 μL Plus reagents (Thermo Fisher Scientific, Waltham, MA, USA) according to the manufacturer’s instructions with minor modifications. The medium and transfection complexes were not removed after transfection. After 24 h, the macrolide antibiotics, 20 μM roxithromycin (Sigma-Aldrich, St. Louis, MO, USA) or 20 μM clarithromycin (Sigma-Aldrich, St. Louis, MO, USA) was given to evaluate whether macrolides could blunt channel function of the mutant *KCNJ5* channel. Calcium channel blocker, 10 μM nifedipine (Sigma-Aldrich, St. Louis, MO, USA), were given to use as downstream inhibitor of *KCNJ5* channel. In addition, the cells were treated with 50 nM angiotensin II (Sigma-Aldrich, St. Louis, MO, USA) for 24 h as a positive control of aldosterone synthesis in HAC15 cells according previous studies [25,26]. After transfection for 72 h, culture supernatant was collected for measuring the concentrations of aldosterone, and cells were harvested for Western blot analysis.

### 2.7. Patch-Clamp Recording

HEK293T cells were seeded into 35-mm dish and transiently transfected with either 2 μg wild-type *KCNJ5* or *KCNJ5* 157-159delITE mutation plasmid plus *KCNJ3* plasmid at 1:1 ratio using 2 μL Lipofectamine 2000 reagent (Thermo Fisher Scientific, Waltham, MA, USA) according to the manufacturer’s instructions. Empty vector was used as a control. Cells were examined at two days after transfection. Standard whole-cell patch clamp recordings were performed using HEKA EPC10 amplifier under the control of Pulse software (HEKA Elektonik Gmbh, Germany). The bath solution was Hank’s balanced salt solution (Gibco; Thermo Fisher Scientific, Waltham, MA, USA) and the pipette solution contained 120 mM aspartate, 5 mM MgCl_2_, 0.1 mM EGTA, 40 mM Hepes, 2 mM ATP, 0.3 mM GTP, pH 7.3. BaCl_2_ (1 mM) was added at the end of the recording to inhibit the current.

### 2.8. Western Blot Analysis

Protein from whole cell extracts were extracted by using RIPA buffer (50 mM Tris base pH 8, 150 mM NaCl, 1% NP40, 0.10% SDS) containing a protease inhibitor (Roche Diagnostics, Indianapolis, IN, USA). The cell lysates were centrifuged and the supernatants mixed with 3X sample buffer (30% glycerol, 15% 2-mercaptoethanol and 1% bromophenol blue). Fifteen micrograms of total protein were loaded to each lane and were separated through 10% SDS-PAGE gels and electrophoretic transferred to PVDF membranes. The membranes were then blocked by incubating in the BlockPRO^™^ blocking buffer (Visual Protein Biotechnology, Taipei, Taiwan) for 1 h and then incubated with blocking buffer containing mouse monoclonal antibody for CYP11B2 (a kind gift from Professor Celso Gomez-Sanchez), anti-KCNJ5 antibody (RRID:AB_10604730, Sigma-Aldrich, St. Louis, MO, USA) and anti-GAPDH antibody (RRID:AB_10167668, Santa Cruz Biotechnology, Dallas, TX, USA) overnight at 4 °C. Following extensive washing in Tris-buffered saline containing 0.1% Tween-20 (TBST) buffer, the transfer membranes were further incubated for 1.5 h in blocking buffer that contained HRP-conjugated secondary antibodies. Then, the membranes were washed three times with TBST. Levels of proteins were detected using chemiluminescent detection reagents (Millipore, Billerica, MA, USA) and visualized using a UVP Biospectrum 810 imaging system (Ultra Violet Products Ltd., Cambridge, UK). Protein expression in each sample was quantified by densitometry using UVP software, normalized to GAPDH levels, and then expressed relative to the control.

### 2.9. Aldosterone Secretion

To measure aldosterone concentration, the culture supernatant was collected 72 h after cell transfection with wild-type control or with *KCNJ5* 157-159delITE mutation. The aldosterone concentrations in culture supernatant were measured by ALDO-RIACT RIA kit (Cisbio Bioassays, Codolet, France). 200 μL of culture supernatant or standards were assayed in aldosterone antibody-coated tubes, and 0.5 mL of ^125^I labeled aldosterone was added to each tube. After incubation for 3 h at room temperature, the contents of the tubes were removed completely. The remaining radioactivity bound to the tubes were measured by a gamma scintillation counter. Aldosterone measurements were normalized using protein concentrations of cell lysates.

### 2.10. Docking Analysis of Roxithromycin with KCNJ5

The homology modeling structures of wild-type and mutant human KCNJ5 protein was generated by the Phyre2 web serve (Structural Bioinformatics Group, Imperial College, London, UK) [27]. To model the KCNJ5 structure, the potassium (K^+^) channel Kir2.2 (PDB ID: 3YJC) was performed as the template structure [28]. The modeling structures were applied for docking stratefgy to identify the binding interaction of KCNJ5 and roxithromycin. The docking analysis was conducted by using the AutoDock 4.2 program (The Scripps Research Institute, La Jolla, CA, USA) with the Kollman charge force field [29,30]. The number of docking poses was set as 100 with default parameters. The decision of the best pose was conducted by free energy of binding and the interaction with highest hydrogen bonds.

### 2.11. Molecular Dynamics Simulation of Roxithromycin with KCNJ5

The molecular dynamics (MD) simulations were carried out using GROMACS v2020.3 to refine the docked conformation [31]. The topology of docked ligand was generated by PRODRG serve [32]. The lipid bilayer is using DPPC (dipalmitoylphosphatidylcholine) derived by Berger, Edholm and Jähnig [33]. The force field for the whole system was GROMOS gromos53a6.ff [34]. The protein–ligand complex was restrained in a box of cubic shape whose edges were placed at 1 nm from the complex of SPC/E water and DPPC model was performed. The system was electrically neutralized by adding 40 Na^+^ ion for wild-type KCNJ5 modeling structure and 36 Na^+^ ion for mutant KCNJ5 modeling structure. The energy minimization was performed using steepest descent and conjugate gradient methods to converge the system to 10 kJ mol^−1^nm^−1^. The system after a short energy minimization step, was performed to NVT (300K) and NPT (1 bar) equilibration with 100 ps running, and LINCS algorithm was used to constrain the hydrogen bond lengths [35]. The final time step MD process was kept 1 ns for the simulation.

### 2.12. Statistical Analysis

Results were expressed as the mean ± SEM. Different results among groups were compared using the Kruskal–Wallis One Way Analysis of Variance on Ranks (ANOVA) or the two-tailed T-test. Statistical analyses are performed with R software and SPSS software. A value of *p* < 0.05 was considered to indicate statistical significance.

## 3. Results

### 3.1. Identification of a KCNJ5 157-159delITE Mutation and Characteristics of the Identified Patient

A *KCNJ5* 157-159delITE mutation in an adrenal adenoma was identified among 228 DNA samples extracted from APA tumoral tissues. A 47-year-old hypertensive woman presented with hypokalemia (3.5 mmol/L) and a 1.2 cm unilateral adrenal mass in the computer tomography. After the standard screening, confirmatory and lateralization test, a unilateral PA over the side of the adrenal mass (APA) was diagnosed. She underwent a laparoscopic adrenalectomy uneventfully and achieved clinical complete success (hypertension-remission without any hypertensive medication) (Appendix A, methods).

We sequenced the DNA extracted from her adrenal tumor and peripheral whole blood and concluded that there was a somatic mutation *KCNJ5* 157-159delITE in this patient (Figure 1A). The deletion of the amino acid sequence from 157 isoleucine (Ile, I), 158 threonine (Thr, T), to 159 glutamine (Glu, E) was localized in the pore loop of the *KCNJ5* channel near the selectivity filter that guides the potassium selectivity (Figure 1B). The evolutionary conservation analysis by BioEdit software 7.2 [36] showed that these three amino acids are conserved in different species (Figure 1C).

### 3.2. Expression of Steroidogenic Enzymes in Human APA Tissues Harboring KCNJ5 157-159delITE Mutation

In the APA harboring *KCNJ5* 157-159delITE mutation, the immunohistochemistry (IHC) analysis showed a strong immunoreactivity and heterogenous expression for aldosterone synthase (CYP11B2) within that adenoma. The IHC staining of 3β-Hydroxysteroid dehydrogenase (HSD3B) was also positive within that adenoma. The immunoreactivity of CYP17A1 in that adenoma did not increase when compared to adjacent normal-looking adrenal tissues. The IHC of CYP11B1was not impressive within that adenoma (Figure 2).

### 3.3. The Electrophysiological Characteristics of the KCNJ5 157-159delITE Mutation

The HEK293T cells were transiently co-transfected with wild-type *KCNJ5* or *KCNJ5* 157-159delITE, together with *KCNJ3*. The cells were voltage-clamped under whole-cell mode with a holding potential of 0 mV and then changed to various potentials to evoke the currents (Figure 3A). For wild-type KCNJ5, the current was outward at the holding potential (0 mV) and became inward when hyperpolarized to −120 and −100 mV. In contrast, the holding current from cells expressing *KCNJ5* 157-159delITE mutant was about 0 pA and became inward when hyperpolarized. The average current amplitude at −120 mV for wild-type *KCNJ5* was −291 ± 73 pA (*n* = 5), which was significantly smaller than that of *KCNJ5* 157-159delITE mutants, −861 ± 295 pA (*n* = 6, *p* < 0.05). Roxithromycin had no obvious effect on the currents of wild-type *KCNJ5* and *KCNJ5* 157-159delITE mutant; neither on the current–voltage relationship. However, the addition of Ba^2+^ greatly suppressed the inward current of wild-type *KCNJ5,* but not that of *KCNJ5* 157-159delITE mutant (Figure 3B); moreover, Ba^2^^+^ also significantly blocked the inward current of the wild-type *KCNJ5* to 149 ± 85 pA at −120 mV (*n* = 5, *p* < 0.01), but showed no effect on the outward current (Figure 3C). These results suggest that *KCNJ5* 157-159delITE deletion enhances the inward current amplitude and concomitantly abolishes the outward current.

### 3.4. Effect of KCNJ5 157-159delITE on CYP11B2 Expression and Aldosterone Secretion

To investigate the effects of the *KCNJ5* 157-159delITE mutation in adrenal cells, HAC15 cells were transfected with wild-type *KCNJ5* or *KCNJ5* 157-159delITE. We observed an increased expression of CYP11B2 in HAC15 cells overexpressing *KCNJ5* 157-159delITE when compared to cells transfected with wild-type *KCNJ5* or empty vector. Clarithromycin, a macrolide antibiotic, could only partially inhibit CYP11B2 expression in the *KCNJ5* 157-159delITE transfected cells. Another macrolide, roxithromycin, could not inhibit CYP11B2 expression in the *KCNJ5* 157-159delITE transfected cells. When nifedipine, a calcium channel blocker, was administrated to cells transfected with *KCNJ5* 157-159delITE mutant, the production of CYP11B2 was decreased. When angiotensin II was given as a positive control, the expression of CYP11B2 was increased significantly (Figure 4A). The aldosterone secretion was increased in culture supernatants of the cells transfected with *KCNJ5* 157-159delITE mutant. Clarithromycin could only partially, while roxithromycin could not, inhibit aldosterone secretion in the *KCNJ5* 157-159delITE transfected HAC15 cells. Besides, nifedipine decreased and angiotensin II stimulated aldosterone secretion in *KCNJ5* 157-159delITE transfected HAC15 cells (Figure 4B).

### 3.5. Docking Analysis and Molecular Dynamics Simulation of Roxithromycin with KCNJ5

The binding-site prediction of macrolide antibiotics to KCNJ5 was according to the similar location identified by Charmandari et al. [37]. The X-ray structure (PDB ID: 3YJC) [28] was set as the template structure for homology modeling. The wild-type and mutant KCNJ5 modeling structures were conducted for docking studies of roxithromycin by AutoDock software [30].

As shown in Figure 5A, it illustrated the binding of roxithromycin in the KCNJ5-L168R through forming the hydrogen bonds with Asn132, Arg155 and Lys160. To further investigate the binding interactions, the predicted binding energy have been carried out by AutoDock software. The result showed that the predicted binding energy of roxithromycin in KCNJ5-L168R was −14.04 kcal/mol (Figure 5A), in *KCNJ5* 157-159delITE it was −4.83 kcal/mol (Figure 5B). As shown in Appendix A, the mutation of L168 to R168 changed the original helix conformation to become a more flexible loop conformation. This flexible loop (Tyr150-Glu159) formed a strong binding to roxithromycin, as shown in Figure 5A; with the Lys160 and Arg155 to support the hydrogen bonds with roxithromycin. However, modeling structure indicated that when part of the loop 157-159 were deleted (in the case of mutation of *KCNJ5* 157-159delITE), roxithromycin lost some interactions with KCNJ5 (Figure 5B). This explained why roxithromycin could not attenuate the hypersecretion of aldosterone in cells transfected with mutation of *KCNJ5* 157-159delITE.

## 4. Discussion

In the present study, we identified a somatic mutation of *KCNJ5* 157-159delITE in our tissue bank of unilateral aldosterone-producing adenomas. The *KCNJ5* 157-159delITE was a gain-of-function somatic mutation. The electrophysiological study demonstrated that the cells transfected with *KCNJ5* 157-159delITE mutation had significantly larger inward current amplitude compared with that of cells transfected with wild-type *KCNJ5*. In the *KCNJ5* 157-159delITE mutant transfected HAC15 cells could increase the CYP11B2 expression as well as aldosterone secretion. Thus, we concluded that the mutant *KCNJ5* 157-159delITE was the determinant factor for aldosterone over-production in APA.

### 4.1. Electrophysiology of Mutant KCNJ5 157-159delITE

Most functional mutations in *KCNJ5*, including the 157-159delITE mutation reported here, are located near or in the selectivity filter (from Thr149 to Gly153) of the channel [37,38]. Some studies proposed that these mutations induced conformational change of the selectivity filter or loop region near the selectivity filter, and led to the *KCNJ5* channel to lose K^+^ selectivity and caused depolarization of the cell membrane; which further activated the aldosterone secretion [37,38].

In whole-cell patch-clamp experiments, HEK293T cells transfected with *KCNJ5* 157-159delITE mutant showed larger inward current and blunted outward currents when compared with that of the cells transfected with wild-type *KCNJ5*. Such results indicated that *KCNJ5* 157-159delITE mutation was a functional mutation from the perspective of electrophysiology. Furthermore, Ba^2+^, which was used as an inhibitor for wild-type *KCNJ5* channel [4], completely blocked the inward current at different voltage levels. However, Ba^2+^ could not alter the inward current at different voltage levels in the cells transfected with *KCNJ5* 157-159delITE or previous reported G151R mutation [4]; even though previously it was reported that Ba^2+^ partially blocked the current of cells transfected with L168R mutation [4].

### 4.2. Expression of Steroidogenic Enzymes in Immunohistochemistry

Aldosterone synthase (CYP11B2) is a key enzyme controls the aldosterone biosynthesis [39]. The immunohistochemistry for CYP11B2 has become an important tool to indicate the excessive production of aldosterone and to identify the location of aldosterone-producing cells in adrenal adenoma or other parts of adrenal tissues [21,40]. The patient harboring somatic *KCNJ5* 157-159delITE mutation showed positive staining for CYP11B2 in her adenoma and in an aldosterone-producing micronodule in the adjacent adrenal tissue. It is classified as classical histopathologic unilateral primary aldosteronism according to the recent updated histopathologic definition [41,42]. It was reported that patients with classical histopathologic findings of unilateral PA have better postsurgical outcomes when compared with that of the nonclassical histopathology group [41]. Indeed, this patient with *KCNJ5* 157-159delITE mutated APA achieved both clinical and biochemical complete success according to PASO consensus criteria [43]. Staining of HSD3B can yield important information of steroid production [44]. Previous study showed positive immunostaining for HSD3B in APA tumors [40]. In this case, the immunohistochemical patterns of HSD3B were similar to those of CYP11B2 in adenoma. The presence of CYP11B1 and CYP17A, the steroidogenic enzymes involved in cortisol production, in adrenal tumor indicate the potential of concomitant autonomous cortisol production [45,46]. Our result showed no immunoreactivity of CYP11B1 and weak immunoreactivity of CYP17A1 presented in this *KCNJ5* 157-159delITE mutated adenoma. Thus, we concluded that the immunohistological data implicated that the adenoma harboring *KCNJ5* 157-159delITE mutation affected the aldosterone production.

### 4.3. Pharmacology and KCNJ5 Mutation

Previous studies showed that unilateral PA patients carrying *KCNJ5* mutations had a better hypertension prognosis after adrenalectomy than that of those non-mutant carriers [20,47,48,49]. Therefore, how to identify whether APA patients harbor *KCNJ5* mutation before surgery is an important topic. Some studies investigated whether certain existing drugs can be used for treatment or differential diagnosis of mutant *KCNJ5*. In 2017, Scholl et al. reported that some macrolide antibiotics, such as roxithromycin and clarithromycin, can specifically alter the electrophysiological pattern of the *KCNJ5* mutant channel and blunt aldosterone production in HAC15 cells transfected with mutated genes, like *KCNJ5* G151R or L168R [16]. Caroccia et al. used the primary culture of aldosterone-secreting cells (CD56^+^ cells) from APA tissues with G151R or L168R mutation to confirm that the macrolide antibiotic, clarithromycin, suppresses *CYP11B2* gene expression and aldosterone production [50]. However, our result suggested that macrolide antibiotic, roxithromycin, did not inhibit the aberrant inward ion current and aldosterone secretion of the *KCNJ5* 157-159delITE transfected cells. We further used docking analysis and molecular dynamics simulation analysis to explore the underlying reason why macrolide is not effective for this mutant. In our study, the complex modeling conformation was taken for further molecular dynamic simulations to let the whole system balance. After 1 ns stimulation the complex structure attained a stable state and equilibrium structure (Appendix A). The evidence showed that roxithromycin blocks the ion channel and influences the function of KCNJ5 L168R mutant due to changing the conformation of flexible loop (Tyr150-Glu159) and helix (Lys160-Leu168) to form strong binding with roxithromycin. In a previous study, Scholl et al. used the structure-activity relationship analysis to characterize the modifications on the lactone ring of macrolide to stabilize the interaction between the macrolides and the mutant channel [16]. Their data also suggested the inhibitory effect of macrolides on mutant KCNJ5 through a direct drug–protein interaction. In our electrophysiological study, wild-type cells treated with roxithromycin was not affected over the inward current at different voltage levels, which was consistent with other study [16]. However, when roxithromycin was administrated to the cells transfected with *KCNJ5* 157-159delITE mutation, the inward current was not inhibited as it was expected in previous reports of cells transfected with *KCNJ5* G151R or L168R mutation [16]. Such data is consistent with the fact that roxithromycin treatment had no significant effect on aldosterone secretion in HAC15 cells transfected with *KCNJ5* 157-159delITE mutation. Because the main interaction of the KCNJ5 channel with roxithromycin comes from the loop structure, the loss of loop amino acids 157-159 in the KCNJ5 157-159delITE channel alters the conformational changes of the channel pore and changes the binding strength to macrolide. As a result, the strong interaction site to roxithromycin in the *KCNJ5* L168R channel is not well preserved and could not be replicated in the KCNJ5 157-159delITE channel. In conclusion, for the diagnosis of *KCNJ5* mutations in PA patients before adrenalectomy, administration of roxithromycin may perform well in patients with *KCNJ5* L168R or G151R mutation but it could not elicit the same result in patients with *KCNJ5* 157-159delITE mutation.

In addition to macrolides, several studies showed that calcium channel blockers, such as nifedipine, can strongly inhibit common mutated *KCNJ5* channel (such as G151R, L168R and T158A)-mediated aldosterone secretion in vitro [51,52,53]. In this study, our result showed nifedipine did also inhibit aldosterone secretion in the cells transfected with *KCNJ5* 157-159delITE mutation. The results indicated that *KCNJ5* 157-159delITE induced aldosterone biosynthesis is dependent on calcium influx. Furthermore, the clinical use of nifedipine might help control blood pressure in such APA/PA patient not only via decreasing aldosterone production associated with *KCNJ5* 157-159delITE mutation, but also via it direct relaxation of vascular smooth muscles.

### 4.4. Limitation and Strength

In this patient, we did not prescribe roxithromycin for her before surgery to investigate whether macrolides or calcium channel blockers could inhibit her aldosterone production before surgery. Due to budget constraints, we can only perform structural analysis on the KCNJ5 channel in the wild-type, L168R mutant and KCNJ5 157-159delITE mutant channels.

To our knowledge, this is the first study using the molecular docking and molecular dynamics simulation analysis to predict the possible interaction site of macrolide with mutant *KCNJ5*. Our result suggested that the three amino acid deletion of KCNJ5 protein, KCNJ5 157-159delITE, caused conformational change of major interaction site for binding with roxithromycin, and resulted in reduced sensitivity to macrolide.

## 5. Conclusions

Our findings suggested that *KCNJ5* 157-159delITE mutation play a significant role in the pathophysiology of aldosterone production. The *KCNJ5* 157-159delITE mutation could disrupt potassium channel selectivity and cause autonomous aldosterone production. Due to the fact that *KCNJ5* 157-159delITE mutation is not sensitive to the inhibitory effects of macrolide, as in the common case of *KCNJ5* L168R and G151R mutations, the expected application of macrolide in diagnosis or treatment of PA/APA patients with *KCNJ5* mutated APA may could have risk of exceptions.

## Figures and Tables

**Figure 1 biomedicines-09-01026-f001:**
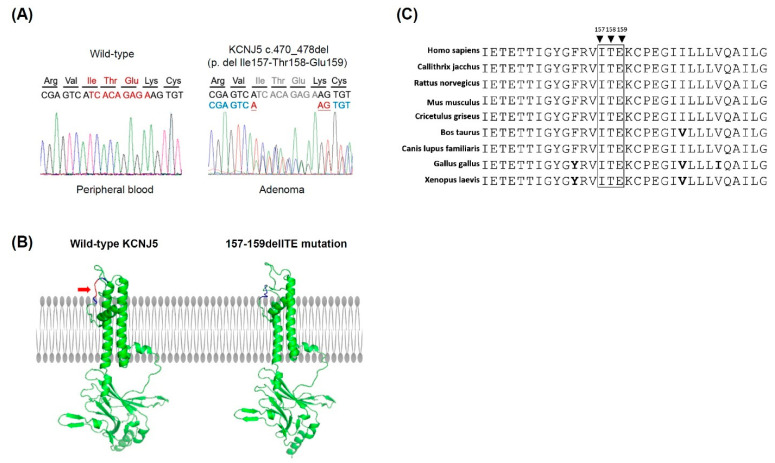
A somatic mutation in *KCNJ5* 157-159delITE in aldosterone-producing adenoma. (**A**) Sequences of blood and tumor genomic DNA of *KCNJ5* in APA. Of note, *KCNJ5* 157-159delITE mutation is noted in adenoma but not blood nuclear cells. (**B**) The modeling structure of human KCNJ5 was generated by the Phyre2 web serve. The red color region marked by red arrow in wild-type KCNJ5 indicates the location of isoleucine (Ile, I) 157, threonine (Thr, T) 158 and glutamine (Glu, E)159. The mutation site of 157-159delITE in KCNJ5 is located in the loop region near the selectivity filter of the potassium channel. (**C**) The evolutionary conservation analysis of amino acids I157, T158 and E159 are conserved in different species.

**Figure 2 biomedicines-09-01026-f002:**
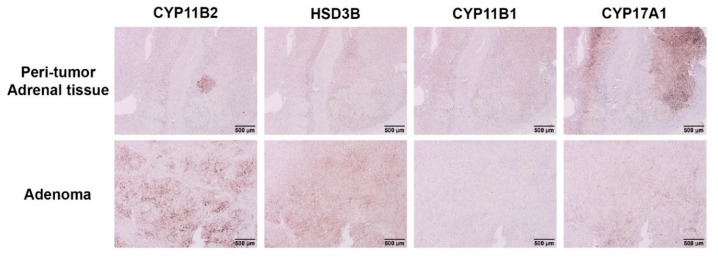
Immunohistochemical staining of CYP11B2, HSD3B, CYP11B1, and CYP17A1 in APA with *KCNJ5* 157-159delITE mutation. Comparing to the peri-tumoral adrenal tissue, the CYP11B2 and HSD3B staining were dominant in *KCNJ5* 157-159delITE mutant adenoma. There was no immunoreactivity of CYP11B1 and weak immunoreactivity of CYP17A1 presented in *KCNJ5* 157-159delITE mutated adenoma. Scale bar, 500 μm.

**Figure 3 biomedicines-09-01026-f003:**
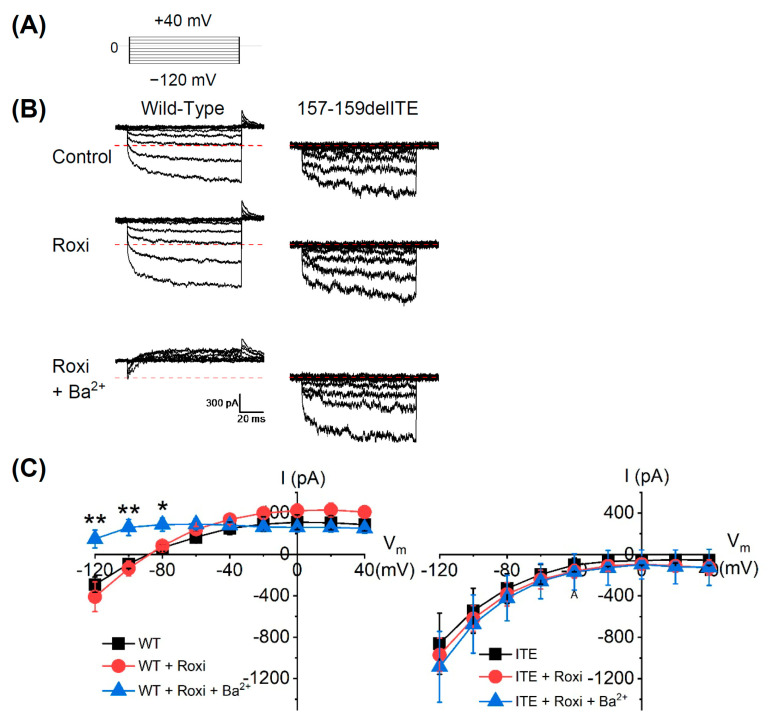
The electrophysiology experiments of *KCNJ5* 157-159delITE mutation. HEK293T cells expressing wild-type (WT) or *KCNJ5* 157-159delITE mutant (ITE) were whole-cell patched under voltage clamp mode with a holding potential of 0 mV. (**A**) The membrane potential stimulation profile. The membrane potentials were changes to various potentials with a step of 40 mV from −120 mV and the currents were recorded. (**B**) The representative current traces from the same cells expressing WT (Left) and ITE (Right). The currents were recorded at 10 min after the formation of whole-cell patch (Control), 5 min after the serial addition of roxithromycin (20 μM, Rox) and Ba^2+^ (1 mM, Rox + Ba^2+^). The red dash lines indicate the zero current. (**C**) The averages current–voltage relationship of cells expressing WT (Left) and ITE mutant (Right). The currents at the last 10-ms interval during the change of membrane potentials were averaged and plotted against the potentials. Data presented were Mean ± SEM; the sample number for WT and ITE were 5 and 6, respectively. * indicated *p* < 0.05 and ** indicated *p* < 0.01.

**Figure 4 biomedicines-09-01026-f004:**
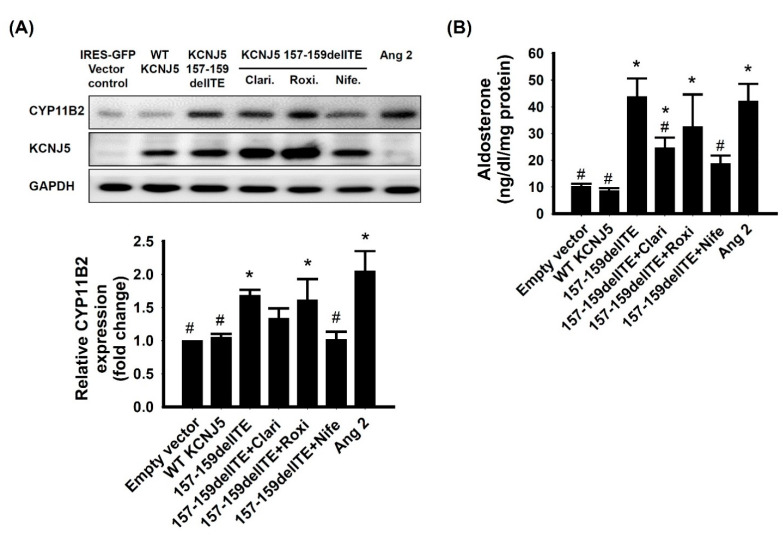
*KCNJ5* 157-159delITE mutation increased CYP11B2 expression and aldosterone production through calcium-mediated pathway. The HAC15 cells were transiently transfected with wild-type or mutant *KCNJ5*. At 72 h after transfection, the samples and culture medium were collected and analyzed. (**A**) The CYP11B2 and KCNJ5 protein expression of cells harboring *KCNJ5* mutant were significantly more than wild-type cells. The macrolides, clarithromycin or roxithromycin did not affect the CYP11B2 expression of cells harboring the *KCNJ5* mutant. (**B**) The aldosterone production of cells harboring *KCNJ5* mutant was significantly more than wild-type cells. The clarithromycin could partially inhibit aldosterone production of the *KCNJ5* mutant cells. However, the roxithromycin could not inhibit aldosterone production of *KCNJ5* mutant cells. Aldosterone production from HAC15 were adjusted by total protein. * indicated *p* < 0.05 vs. wild-type *KCNJ5* group, # indicated *p* < 0.05 vs. 157-159delITE group. The macrolides were used as the inhibitors of mutant KCNJ5 channel. Nifedipine was used as inhibitor of downstream of *KCNJ5*-related pathway. Angiotensin II was a stimulator of aldosterone production.

**Figure 5 biomedicines-09-01026-f005:**
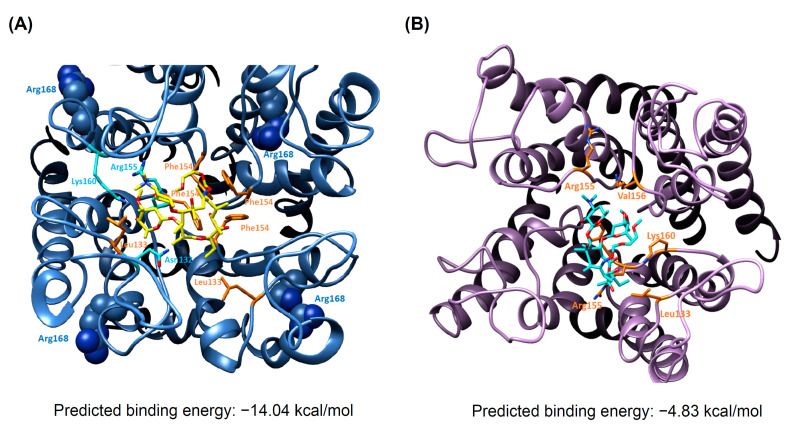
Docking analysis and molecular dynamics simulation of roxithromycin with mutant KCNJ5. (**A**) The binding site analysis for Roxithromycin (Cyan) with L168R mutation of KCNJ5 after 1 ns MD equilibrium. Roxithromycin forms hydrogen bonding with Asn132, Arg155 and Lys160 (as shown in orange color residues). The hydrophobic effect was contributed by Leu133, and Phe154 (as shown in orange color residues). (**B**) The binding site analysis for Roxithromycin (Cyan) with KCNJ5 157-159delITE mutation of KCNJ5 after 1 ns MD equilibrium. Roxithromycin forms weak hydrophobic and hydrogen bond interaction with Leu133, Arg155, Val156 and Lys160 (as shown in orange color residues).

## Data Availability

All data are present in the manuscript or available by request to corresponding author, V.C.W. (q91421028@ntu.edu.tw).

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
