# Peer review of "Pathophysiological and Pharmacological Characteristics of KCNJ5 157-159delITE Somatic Mutation in Aldosterone-Producing Adenomas"

_biomedicines, 2021, doi:10.3390/biomedicines9081026_

Round 1

Reviewer 1 Report

The authors detected a novel somatic mutation of KCNJ5 and demonstrated its electrophysiological and functional aspects based on in vitro experiments. The context is well-written, but some points should be discussed or modified in the manuscript.

  1. Please describe about the ethics approval for this study, including DNA analysis.
  2. The authors concluded that this novel somatic mutation does not influence cortisol secretion (LINE 376-380), whereas previous studies demonstrated KCNJ5-mutated APAs harbor cortisol co-secretion more frequently than the other APAs due to intra-tumoral CYP11B1 expression (PMID: 30020487 etc.). Do the authors have any data on in vitro CYP11B1 expression or cortisol co-secretion in this case? Is there any difference between this novel mutation and known ones (G151R and L168R)? If the authors conclude as above, those data should be shown in the text.
  3. For figure 4, the WT KCNJ5 group seems appropriate for controls to compare with the KCNJ5 157-159delITE groups.

Minor comments

-RRID for each antibody used in this study should be provided in the method section.

-Please refine the layout of supplemental table 1 for easier understanding about biochemical changes between before and after adrenalectomy.

-In LINE 364, "somatic" could be inserted prior to KCNJ5 157-159delITE mutation.

Reviewer 2 Report

Authors found a new somatic mutation in KCNJ5 gene in APA patients. They found KCNJ5 157-159delITE mutation and it showed loss of potassium ion selectivity and induced autonomous aldosterone production. Furthermore, they demonstrated that macrolide antibiotics could not interfere the aberrant electrophysiological properties.

 This is interesting and valuable study as it provides insights into the structure and function of KCNJ5 gene, however I have some question, which should be clarified.

Page 3, line 115, “mutant KCNJ5 transfection and drugs treatment” in materials and methods.

Authors should describe transfection method in detail because transfection experiment system for adrenal cell line has problem of transfection efficiency.

Specifically, authors noted that “the HAC15 cells were seeded into 6-well plates at a density of 1×106 cells/well.” Did they use collagen-coated plate or not? Some adrenal cell lines showed difficulties in attachment to plate.

Authors used lipofection with “Lipofectamine® LTX reagents” for plasmid DNA transfection of both HAC15 cells and HEK293T cells. Please check company name because “Life Technologies, Carlsbad, CA, USA” and “Thermo Fisher Scientific, Waltham, MA, USA” were used for this reagent in this article. Did authors also use “PLUS™ Reagent” when they use LTX reagent?

For lipofection of plasmid into HAC15 cells, please provide transfection efficiency because lipofection for HAC15 cells showed low efficiency and some researchers choose electroporation. Authors transfected plasmids including EGFP, therefore they can easily show the results of transfection efficiency by images using fluorescence filter. I also requested that qPCR data of KCNJ5 gene to prove the successful transfection.

Please describe the amount of plasmid DNA is transfected into HAC15 cells? 1μg for each 6 well plate?

I assume that results of lipofection with HAC15 cells were show in figure 4, but legends described that “The HAC15 cells were transiently electroporated”. Did authors used electroporation? If so, there was no description about electroporation system.

Regarding Fig 4, authors used antibiotics and angiotensinâ…¡ for 48 hours. Because angâ…¡ stimulates adrenal cell line quite rapid time course, observation after 48hrs from addition seemed to be inappropriate. As peak of angâ…¡ may be ranged in 6~12hrs, experiment system should be considered again or describe carefully why authors choose 48hrs. When authors think 48hrs are appropriate to show enough upregulation of CYP11B2 in HAC15 cells, I recommend authors show time course upregulation of CYP11B2 by angâ…¡ in accordance with 0, 3, 6, 12, 24, 48hrs.

Page 4, line 146, “western blot analysis” in materials and methods. Please describe the amount of protein samples used for one lane.

Page 4, line 165, “Aldosterone secretion” in materials and methods. Authors described that ALDO-RIACT RIA kit for detection of aldosterone. I assume that authors used “aldosterone radioimmunoassay kit” from Cisbio bioassays. As this kit is usually used for urine, serum and plasma human samples, authors should describe carefully how they used this kit. For example, when authors collected culture supernatant after 72 hours from lipofection, 1) What is the scale of transfection (did they use 6-well plate? What is the amount of medium? 2ml?) 2) Did they change medium after transfection? If not, were lipofection reagents and plasmids included in medium for measurement? 3)How much medium was used for measurement? Whole supernatant was used for measurement? Or, only a part of medium was used for making dilution series for measurement?

Regarding Fig 4, I recommend authors show the upregulation of CYP11B2 and inhibition by macrolides for KCNJ5 L168R plasmid transfection as positive control of this experiment system. I am afraid that uniqueness of KCNJ5 157-159delITE is not proved if same results were appeared when they used KCNJ5 L168R plasmid in their experiment system.

Round 2

Reviewer 2 Report

Authors replied my questions appropriately.